# COVID-19 Pandemic: 1-Year Follow-Up in Children and Adolescents with Neuropsychiatric Disorders

**DOI:** 10.3390/ijerph20053924

**Published:** 2023-02-22

**Authors:** Grazia Maria Giovanna Pastorino, Marianna Marino, Salvatore Aiello, Raffaella D’Auria, Rosaria Meccariello, Antonietta Santoro, Andrea Viggiano, Francesca Felicia Operto

**Affiliations:** 1Dipartimento di Medicina, Chirurgia e Odontoiatria “Scuola Medica Salernitana”, Via S. Allende, 84081 Baronissi, Italy; 2Dipartimento di Scienze Motorie e del Benessere, Università di Napoli Parthenope, Via Medina 40, 80133 Napoli, Italy

**Keywords:** COVID-19, neuropsychiatric disorders, children, adolescents, emotional behavioral symptoms, parental stress, follow-up

## Abstract

Introduction: Few studies have focused on the long-term effects of the COVID-19 pandemic on mental health. The objective of our work was to evaluate the changes in emotional and behavioral symptoms in patients with neuropsychiatric disorders and the impact on parenting stress 1 year after the first national lockdown. Methods: We enrolled 369 patients aged 1.5–18 years of age referred to the Child and Adolescent Neuropsychiatry Unit of the University Hospital of Salerno (Italy) by their parents. We asked their parents to complete two standardized questionnaires for the assessment of emotional/behavioral symptoms (Child Behavior CheckList, CBCL) and parental stress (Parenting Stress Index, PSI) prior to the pandemic (Time 0), during the first national lockdown (Time 1) and after 1 year (Time 2), and we monitored the changes in symptoms over time. Results: After 1 year from the start of the first national lockdown, we found a significant increase of internalizing problems, anxiety, depression, somatization, and social and oppositional-defiant problems in older children (6–18 years), and a significant increase of somatization, anxiety problems, and sleep problems in younger children (1.5–5 years). We also observed a significant relationship between emotional/behavioral symptoms and parental stress. Conclusion: Our study showed that parental stress levels increased compared to the pre-pandemic months and continues to persist over time, while internalizing symptoms of children and adolescents showed a significant worsening during 1 year follow-up from the first COVID-19 lockdown.

## 1. Introduction

The World Health Organization (WHO) proclaimed the epidemic of the new coronavirus SARS CoV-2 (COVID-19) a public health emergency of international concern in January 2020, following an outbreak of pneumonia of unknown cause in the Chinese city of Wuhan. The WHO labeled the coronavirus illness (COVID-19) a global pandemic on 11 March 2020 [1]. To address the emergency at the national level, the Council of Ministers in Italy issued several strict restrictions, such as the closing of all schools and the quarantine of all Italian citizens. The strictest restrictive measures were in force from March to May 2020 but, to deal with the second wave of COVID-19, containment measures for different scenarios (the establishment of the red, orange and yellow zones) were applied from November 2020 to March 2021. The outbreak of the COVID-19 health emergency during these last years has raised numerous moral and health-related issues regarding how to effectively respond and stop the spread of the virus. Public health initiatives have concentrated on adopting measures such as quarantines that benefit the larger community to control the pandemic, without neglecting the need to acknowledge the importance of human rights, especially concerning those individuals living with a disability [2]. In fact, social isolation caused by the quarantine has been shown to have a deleterious impact on the emotional well-being of the general population, despite being successful in controlling the virus [3]. According to Hawryluck et al. [4], prolonged isolation has been linked to increased post-traumatic stress disorder (PTSD) symptoms: this could imply that the quarantine can be experienced as a specific trauma. Moreover, as found by Lee and colleagues [5], the COVID-19 pandemic and the resulting self-isolation had a significant impact on people’s everyday and social lives, and an increase in the occurrence of symptoms such as mood deflection, irritability, and insomnia resulted. It has been highlighted both the quarantine itself and its duration are associated to stress factors such as fear of infection, insufficient resources and information, and social stigma, which increased in the period following the lockdown [6]. An Italian online survey conducted on 213 young adults during the first lockdown suggested that social connectedness and loneliness were significant predictors of quality of life (QoL). The authors highlighted the importance of analyzing both social context and psychological factors in order to devise intervention strategies to improve the QoL during COVID-19 lockdown, and underlined how much human relationships are fundamental for maintaining physical and psychological wellbeing [7]. It is evident that personal and environmental factors influence the mental and emotional response to an outbreak [8]. Specifically, a Chinese study examined a sample of psychiatric patients to determine the psychopathological impact during the COVID-19 lockdown. Psychiatric patients showed greater levels of PTSD, depression, anxiety, stress, insomnia, concerns about their physical health, anger and irritability, and suicidal ideation compared to controls [9]. Furthermore, the lockdown posed a serious risk to children and teenagers, since it caused the restriction of activities and relationships that are essential for their growth. Clinging, inattention, and irritation were the most important behavioral issues found by Gritti et al. [10]. According to a study conducted in late 2020 [8], the increase in the prevalence of anxious and depressive symptoms in children and adolescents was linked to the pandemic itself, social isolation, and parental stress. Furthermore, according to several studies done on the paediatric population, the dramatic change in daily routines exacerbated emotional symptoms and self-regulation issues [11,12]. The alterations of the daily routine have had a negative impact on the psychopathological well-being of parents and children [13,14], as highlighted in several Italian studies that have shown the adversities experienced by mothers and their children during the national lockdown [11,12]. A recent study conducted in India found that many children experienced an increase in symptoms of anxiety and depression during the 15 months of the pandemic: the extent of this increase was associated with various vulnerability factors such as developmental age, educational status, pre-existing mental health condition, or quarantine due to infection/fear of infection [15].

Loneliness and lack of routine had a greater effect on those young people already suffering from serious mental health conditions, such as autism spectrum disorder, psychosis, or anxiety disorders [16]. In this regard, during the first and second wave of the COVID-19 pandemic, the Child and Adolescence Neuropsychiatry Service at the Children’s Hospital Bambino Gesù in Rome recorded a significant increase in access for mood disorders, self-injury behaviors, and suicidal ideation [17]. As reported by Guessom et al. [18], adolescents with psychiatric disorders were particularly vulnerable given the interaction between care interruption, COVID-19 related anxiety, and problems adjusting to confinement. Nevertheless, as found by Conti and colleagues, Italian children with psychiatric disorders responded to quarantine better than children with neurological or neurodevelopmental issues: the existence of internalizing psychiatric disorders could be connected to a “protective role” of the quarantine, due to the lowering of various stressors (going to school or competition with peers) [19]. Furthermore, it has been noted that children and adolescents with neurodevelopmental disorders and/or externalizing disorders have been the ones most impacted by the implemented restrictive measures [3]. In addition to these data, as reported by studies conducted during 2021 and 2022 among children with autism spectrum disorder (ASD) and their families, parents of children with ASD exhibited higher levels of psychological distress, highlighting the increase in anxiety and depression problems [20,21]. Regarding Attention-Deficit/Hyperactivity Disorder (ADHD), it has underlined an exacerbation of inattention and hyperactivity symptoms during the COVID-19 pandemic, in addition to stress and mood alterations [22]. Furthermore, according to research conducted during 2022 [23], the epilepsy course in paediatric patients appeared to be steady during the COVID-19 pandemic, while children’s behavioral changes and sleep issues could be connected to an increase in seizure frequency. Undoubtedly, the lack of a regular schedule could affect such factors [23]. In addition, as highlighted during a study among Jordanian children [24], COVID-19 pandemic measures have affected children’s behavior and emotional state, underlining a more irritable mood, a greater likelihood to argue with the rest of the family, and a decrease in physical activity and hours of sleep. Considering children and adolescents with behavioral disorders, the exacerbation of some behavioral issues due to the restrictions was found. The insistence on sameness, the disruption of routines, the uncertainty surrounding the emergency, and the suspension of individual and group rehabilitation interventions may have had a role in increasing behavioral problems, as well as in the increase in challenges faced by parents in controlling their children’s attitude [3]. On the other hand, as found by Samji et al. [25] increasing supports for children, adolescents, and families, as well as the implementation of preventive measures, may reduce long-term mental health consequences in children and adolescents. A significant positive correlation between children internalizing symptoms and parental distress has been found [26], pointing out the need to support parents in stressful situations when their children present internalizing symptoms. Recent studies conducted on children and adolescents with [3,26,27,28,29,30] and without [13,31] neuropsychiatric disorders suggested a relationship between the emotional/behavioral symptoms of children and parental stress. Ueda et al. [32] compared the quality of life (QOL) of parents and children with neurodevelopmental disorders before and one year after the COVID-19 pandemic, underlining the connection between children’s quality of life and parental stress in the time following the pandemic. It has been highlighted that the COVID-19 pandemic had an important impact on the emotional and mental well-being of families. Parents have had to deal with their children’s problems without outside help, such as from school or therapy, and this has made them more concerned and has diminished their sense of effectiveness [32]. The quality of life and mental health of parents and children with neurodevelopmental disorders are likely to change in the future, affecting one another and being influenced by societal factors and public assistance at the same time [32].

As reported, although there are many studies in the literature on the impact of COVID-19 on the mental well-being of patients with neuropsychiatric disorders and their families [3,17,26,27,28,29,30,33], to the best of our knowledge there are no follow-up studies that have evaluated long-term sequelae.

In light of this, the primary goal of this study is to examine changes in the emotional and behavioral characteristics of Italian adolescents and children who had been referred for neuropsychiatric disorders during and following the implementation of the first national lockdown, as well as how these changes affected parental stress. Specifically, attention has been focused on the development of these alterations in a subsequent setting by considering the circumstances prior to the pandemic (T0), throughout it (T1), and a year (T2) after the initial nationwide lockdown.

## 2. Materials and Methods

### 2.1. Participants

Our study was conducted at the Child and Adolescent Neuropsychiatric Unit of the University Hospital in Salerno (Italy), a complex operative unit for the diagnosis and pharmacological treatment of children and adolescents with neurological and psychiatric disorders (0–18 years).

We enrolled all patients aged 1.5–18 who referred to our clinic in the months preceding the pandemic (September 2019–January 2020—Time 0) and whose parents had completed two standardized questionnaires for the assessment of emotional/behavioral symptoms: the Child Behavior CheckList 6–18 years, (CBCL), and the Parenting Stress Index–Short Form (PSI/SF), as is our usual clinical practice.

All of the participants had received a previous neuropsychiatric diagnosis in our unit between 2018–2019 from a multidisciplinary team (child neuropsychiatrists, speech therapists and child psychologists) based on DSM-5 criteria [34] and supported by clinical observations, standardized neuropsychological tests, and instrumental investigations.

The parents of all recruited patients were remotely contacted during the first lock-down (March–May 2020—Time 1) and 1 year after the first lockdown (March–May 2021—Time 2) and were asked to repeat the CBCL and PSI/SF questionnaires.

A detailed explanation about the purpose and the procedures of the study was provided to all parents who sent their informed consent in written form by e-mail. The only exclusion criterion was the poor compliance of the parents.

A team composed of six child psychologists and two child neuropsychiatrists analyzed the data collected.

The comparison between Time 1 and Time 0 was already the object of our previous study [33]; in the present protocol we focused on the comparison between Time 2 and Time 1 in order to monitor symptoms after a year of the pandemic.

In our analysis, we considered gender, age, diagnosis, comorbidities, drug therapy, age, and level of education of the parents (years of education).

The study was approved by the “Campania Sud” Ethics Committee (protocol 61902) and followed the guidelines of good clinical practice.

### 2.2. Child Behavior CheckList

The Child Behavior Checklist (CBCL) [35] is a standardized questionnaire for parents that is designed to assess the emotional and behavioral symptoms in children and adolescents up to 18 years of age. There are two versions compiled by the caregivers: one is used for children from 1.5 years to 5 years old and another for those from 6 to 18 years old. Each version includes 113 questions, and there are three possible answers, based on the following Likert scale: 2 = true or very true; 1 = sometimes true; 0 = false. The raw scores were converted into t-scores based on gender and age. The t-scores were distributed among six DSM oriented subscales and eight empirical subscales that can lead to the obtaining of three main scales: Externalizing, Internalizing and Total Problems.

We can interpret the t-score of the DSM-oriented subscales and empirical subscales as follows: a t-score ≤ 64 is normal, a borderline range is indicated by a t-score between 65 and 69, and a t-score ≥ 70 indicates clinical symptoms.

Moreover, we can interpret the t-scores of the main scales based on the following interval: a t-score ≤ 59 indicates normal scores, a t-score between 60 and 64 indicates a score that is within a borderline range, and clinical symptoms are indicated by a t-score ≥ 65.

The CBCL has an adequate internal consistency [36]. Alpha reliability coefficients based on the responses of individuals in the normative sample ranged from 0.66 to 0.95 for the 1.5–5 years subscales, and from 0.72 to 0.97 for the 6–18 years subscales. The test-retest reliability score ranged from 0.68 to 0.92 for the 1.5–5 years subscales, and from 0.82 to 0.94 for the 6–18 years subscales.

### 2.3. Parental Stress Index

The PSI Short Form (PSI/SF) [37] is a standardized questionnaire for parents that consists of 36 sentences. The parents/caregivers can answer to each sentence according to a 5-point Likert scale, ranging from “strongly agree = 5” to “strongly disagree = 1”. The PSI/SF provides three subscales: Parental Distress (PD), which represents the stress related to the parenting role, the Dysfunctional Parent-Child Interaction Scale (P-CDI), which assesses the level of stress related to parent-child interaction, and the Difficult Child Scale (DC) which assesses the stress related to the characteristics of a child [36]. Finally, from these three subscales it is possible to provide a main scale which measures the levels of Total Stress (TS).

In PSI/SF the raw score is converted to t-scores; a t-score ≥ 85 indicates clinically significant parental stress [38].

Alpha reliability coefficients based on the normative sample ranged from 0.78 to 0.88 for Child Domain subscales and from 0.75 to 0.87 for Parent Domain subscales. Reliability coefficients for the Total Stress scale were 0.96 or greater, indicating a high degree of internal consistency. Test-retest reliability coefficients ranged from 0.55 to 0.82 for the Child Domain, from 0.69 to 0.91 for the Parent Domain, and from 0.65 to 0.96 for the Total Stress score.

### 2.4. Statistical Analysis

We expressed all the neuropsychological scores as mean ± standard deviation (SD). For evaluation of the changes in mean scores over time, we performed the ANOVA analysis of variance for repeated measures by a within-subject and inter-subject test. The subsequent post-hoc analysis was performed by a Bonferroni test.

The possible relationship among different data was explored through the Spearman’s correlation test. We considered statistical significance to be a *p*-value < 0.05.

Statistical Package for Social Science software, version 23.0 (IBM Corp, 2015, Armonk, NY, USA) was used for our statistical analysis.

## 3. Results

### 3.1. Sample Characteristics

At Time 1 the total sample included 383 families. At Time 2, 14 of these (3.7%) had chosen not to participate. The children were aged between 2–18 years. The total sample included children and adolescents with the following neuropsychiatric diagnoses: autism spectrum disorder, epilepsy, specific learning disorders, intellectual disability, communication disorders, attention deficit/hyperactivity disorder, behavioral disorders, anxiety disorders, and mood disorders. In Table 1 and Table 2 the main socio-demographic and clinical characteristics are summarized.

### 3.2. Comparison of Mean Scores of PSI and CBCL during the First Lockdown and 1-Year after the Pandemic

The analysis of variance showed significant changes over time in all the subscales of PSI and CBCL 6–18; in the CBCL, 1.5–5 significant changes were recorded in all the subscales considered, except for the attention problems subscale (*p* = 0.063).

The post hoc analysis showed that the comparison between mean PSI scores during the first lockdown (Time 1) and 1 year after the pandemic (Time 2) showed no significant changes in all subscales analyzed in the total sample (*p >* 0.05 in all the PSI subscales).

The mean scores of the following subscales of CBCL 6–18 were statistically increased at Time 2 compared to Time 1: Anxiety/Depression (*p* < 0.001), Withdrawal/Depression (*p* = 0.017), Somatic complaints (*p* = 0.008), Socialization (*p* = 0.005), Affective problems (*p* < 0.001), Anxiety problems (*p* < 0.001), Somatic problems (*p* = 0.024), Oppositional-defiant problems (*p* = 0.012), Internalizing problems (*p* < 0.001), and Total problems (*p* < 0.001). We did not find significant changes in the remaining subscales.

The mean scores of the following subscales of CBCL 1.5–5 were statistically increased at Time 2 compared to Time 1: Somatic complaints (*p* = 0.043), Sleep problems (*p* = 0.012), and Anxiety problems (*p* = 0.024).

We summarized the mean PSI and CBCL scores and the statistical comparison in Table 3 and Table 4 and Figure 1.

### 3.3. Correlation Analysis between PSI and CBCL Subscales

The correlation analysis showed significant positive relationship between the PSI scales Total Stress (TS), Parental Distress (PD), Parent-Child Difficult Interaction (PCD-I), Difficult Child (DC), and the CBCL scales Total Problems, Internalizing Problems and Externalizing Problems. All of the results of the correlation analysis are summarized in Table 5.

## 4. Discussion

The aim of our study was to evaluate the long-term mental health impact of the COVID19 pandemic on an Italian population of children and adolescents with neurological and psychiatric disorders comparing the first lock-down period (March–May 2020—Time 1) with 1 year after the first lockdown (March-May 2021—Time 2). We focused on psychological, emotional, and behavioral changes and the effect on parental stress before (Time 0), during (Time 1), and 1 year after (Time 2) the first national lockdown. Since the comparison between Time 0 and Time 1 was analyzed in one of our previous studies [33], we present here the results of the comparison between Time 1 and Time 2 in order to monitor symptoms after one year of the pandemic. While there are many studies comparing the mental health status of Italian neuropsychiatric patients before and during the first lockdown [3,19,22,30,39], to the best of our knowledge, our study is the first that has attempted to assess the long-term effects of the COVID-19 lockdown on mental health. Our sample included 369 patients and their families (Table 1 and Table 2), aged between 2–18 years, with the following neuropsychiatric diagnoses: autism spectrum disorder (*n* = 110), epilepsy (*n* = 90), specific learning disorders (*n* = 40), intellectual disability (*n* = 33), communication disorders (*n* = 29), attention deficit/hyperactivity disorder (*n* = 21), behavioral disorders (*n* = 16), anxiety disorders (*n* = 15), and mood disorders (*n* = 15). All of the parents completed two standardized questionnaires for the assessment of emotional/behavioral symptoms of the patients (CBCL) and of parental stress (PSI) during and 1 year after the first national lockdown.

Overall, our results showed that the lockdown had an impact on the well-being of children and adolescent with neurodevelopmental disorders and their families [33] and that, in the one-year follow-up, this impact had further effects.

In greater detail, regarding the CBCL 6–18, the statistical analysis of the changes in the mean scores over time showed that there was a significant worsening in all subscales analyzed during the first lockdown compared to the pre-pandemic period; furthermore, after the 1 year follow-up, the following scales had worsened: Anxiety/Depression, Withdrawal/Depression, Somatic complaints, Socialization, Affective problems, Anxiety problems, Somatic problems, Oppositional-defiant problems, Internalizing problems, and Total problems. These findings suggest that internalizing problems, such as anxiety and depression, had a greater impact on school-age children and adolescents. As mentioned, there are no comparable results in the literature. However, there are follow-up studies that have analyzed the long-term impact of the pandemic in the general pediatric population and in patients without neurological or psychiatric diagnoses [15,40]. Pustake et al. [15] compared levels of anxiety and depressive symptoms between February–March 2020 and May–July 2021 in a population of 1255 children and adolescents aged 6–16 without any psychiatric or chronic disorders. They found an increase in both anxiety levels and symptoms of depression [15]. These results suggest that, in a general pediatric population, internalizing symptoms increased during one year of follow-up. It is known in the literature that patients with pre-existing psychiatric diagnoses have suffered the most serious consequences of the lockdown [41]. It is possible to hypothesize that, although there has been an easing of restrictions, the containment measures applied by the Italian government between November 2020 and March 2021, (including distance learning for secondary school and the closure of all recreational and sporting activities) have had a further impact on internalizing symptoms. Conversely, we did not find significant changes in the externalizing problems: this could be due to parents’ greater attention to internalizing problems, after an initial concern about externalizing behaviors [33]. It must be considered that during the second wave, patients were able to carry out neuro-psychomotor treatment regularly, contrary to what happened during the first months of the pandemic: the usefulness of neuro-psychomotor treatment in children with neuropsychiatric disorders was recognized, in particular on behavioral symptoms [42]. Furthermore, externalizing problems are reported to be predictors of internalizing problems over time [43,44] and, considering our 1 year follow-up, this factor may have affected our results.

Regarding the CBCL 1.5–5, the analysis of the changes over time showed a significant worsening in all subscales during the first lockdown compared to the pre-pandemic period, while, after 1-year follow-up, the following scales were further worsened: Somatic complaints, Anxiety problems, and Sleep problems. Again, although there are some studies comparing results before and during the lockdown [45,46,47], we did not find as many follow-up data. However, despite the lack of other data in the literature, our experience suggests that anxiety symptoms and somatization are still high after 1 year of the pandemic, even in younger children. In the sample of children under the age of 6, sleep disturbances had also worsened further, probably due to the change in habits and anxiety symptoms. Regarding sleep disturbances, the English National survey follow-up on mental health of the youth population showed an increase in sleep disturbances during the second wave of the COVID-19 (spring 2021), especially in patients with mental disorders [40].

It is already known that neurodevelopmental disorders are related to an increase in parental stress levels [48,49,50,51]. The analysis of change in parental stress over time showed a significant increase in parental stress levels during the first lockdown compared to pre-COVID levels; on the contrary, after 1-year follow-up no significant change in parental stress was found. This could suggest that caregivers of patients with neuropsychiatric diagnosis were already stressed during the first lockdown and are maintaining their stress levels.

A correlation analysis showed a significant relationship between parental stress levels and both internalizing and externalizing symptoms of children. In this regard, Cusinato et al. [13] demonstrated that the resilience of the whole family has a mutual influence on mental well-being and the ability to successfully cope with changes, both in parents and children: it follows that the levels of stress and depression in children are correlated with those of the parents [13]. Similar results are reported by a recent Italian cross-sectional study [26] on pediatric patients with neuropsychiatric pathology: the authors found, during the lockdown, a correlation between CBCL Internalizing problems of all PSI subscales, and between the CBCL Externalizing problems and Difficult Child subscales.

To the best of our knowledge, this is the first follow-up study on the long-term effect of the COVID-19 lockdown on the mental health of the Italian population of children and adolescents with various neuropsychiatric disorders. The strengths of our study are the use of standardized quantitative questionnaires, the focus on a specific population of patients (adolescents with neuropsychiatric disorders), and the presence of pre-existing initial study results [33] and the long-term follow-up. However, this study has some limitations, the first of which is the lack of a control group that would have allowed us to compare the evolution of emotional-behavioral symptoms in our sample and in a sample of children and adolescents without neuropsychiatric conditions. Another limitation of the study is that some social and psychological factors could have affected the emotional-behavioral profile and parental stress (such as social connection, social support, feeling of loneliness, fear of contagion). Furthermore, our study was conducted on a population homogeneous by geographical origin, therefore our conclusions are strictly related to the local conditions in the selected time (containment measures adopted, level of contagion, etc.). Another limitation of the study is that we considered in two large ranges of age (under and over 6 years) in our analysis, so it will be suitable to divide the sample into smaller age groups, making a comparison between them, and this will be the aim of a future study.

In conclusion, our research showed a worsening of internalizing symptoms in both children and adolescents with neuropsychiatric conditions during 1 year follow-up from the first COVID-19 lockdown, which could suggest that the pandemic and the containment measures adopted have had a more evident prolonged effect on internalizing symptoms. Our study also showed that parental stress levels, after an initial increase during the first lockdown compared to the pre-pandemic months, still remains high after 1 year of the pandemic, and it is related to the internalizing/externalizing symptoms of their child. These findings are a call for further studies of long-term sequelae on the mental well-being of neuropsychiatric patients and their families.

## Figures and Tables

**Figure 1 ijerph-20-03924-f001:**
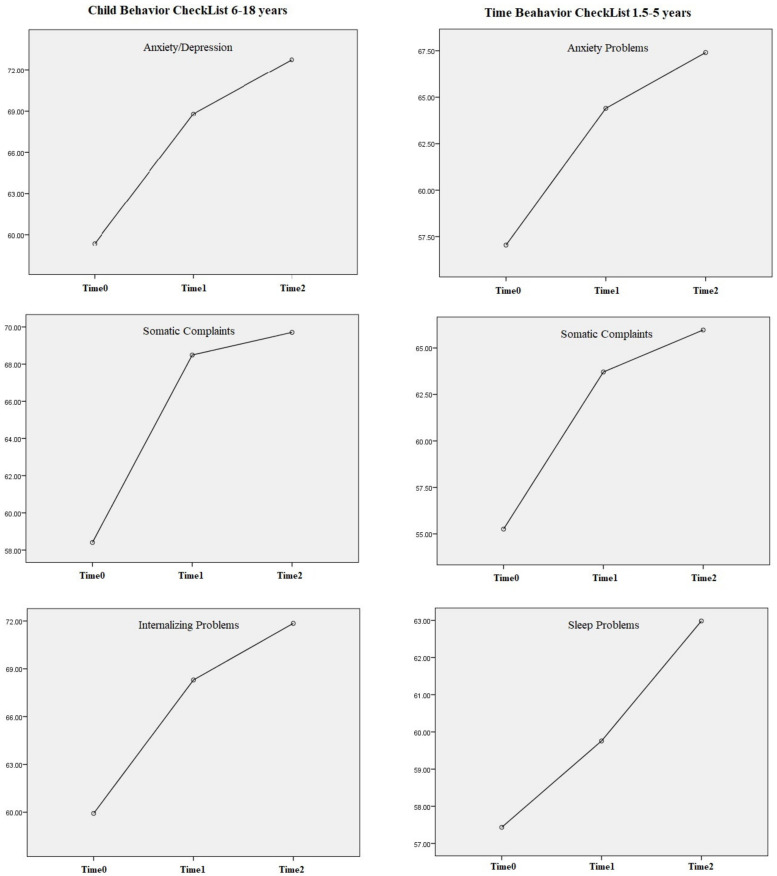
Significant changes in mean scores over time of Child Behavior CheckList.

**Table 1 ijerph-20-03924-t001:** Main socio-demographic characteristics of the families included in our study at Time 1 and Time 2.

Participants	Time 1 *n* = 383	Time 2 *n* = 369
child age (mean ± SD) (years)	9.89 ± 4.42	9.84 ± 4.29
sex		
male	233 (61%)	221 (60%)
female	150 (39%)	148 (40%)
father age (mean ± SD) (years)	43.87 ± 7.05	43.92 ± 7.09
mother age (mean ± SD) (years)	40.82 ± 6.34	40.82 ± 6.29
maternal education level (mean ± SD) *	14.07 ± 3.89	13.99 ± 3.94
paternal education level (mean ± SD) *	14.07 ± 3.71	14.02 ± 3.75

SD = standard deviation. * Calculated in years of school.

**Table 2 ijerph-20-03924-t002:** Main socio-demographic and clinical characteristics of the total sample and of the sub-samples divided by main neuropsychiatric diagnosis at Time 2.

Principal Diagnosis *	Age (Years)	Sex	Father Age (Years)	Mother Age (Years)	Characteristics of the Principal Disease	Neuro-Psychiatric Comorbidities	Other Clinical Conditions	Drug Therapy
**autism spectrum disorder** ***n* = 110**	8.1 ± 3.92	male = 75 (68%)	45.18 ± 6.62	41.75 ± 6.08	level 1 = 41 (37%) level 2 = 43 (39%) level 3 = 26 (24%)	34 (31%)	28 (25%)	21 (19%)
**epilepsy** ***n* = 90**	12.35 ± 4.25	male = 50 (56%)	45.85 ± 7.40	42.38 ± 7.15	focal = 49 generalized = 30 unknown = 10	24 (27%)	16 (18%)	81 (90%)
**specific learning** **disorders** ***n* = 40**	10.50 ± 2.30	male = 21 (53%)	40.65 ± 4.86	38.50 ± 4.04	mixed = 28 (70%) dyslexia + dysorthography = 9 (23%) only dyscalculia = 3 (8%)	7 (18%)	6 (15%)	0 (0%)
**intellectual** **disability** ***n* = 33**	8.70 ± 4.97	male = 18 (55%)	43.72 ± 6.19	40.79 ± 6.39	mild = 20 (61%) moderate = 9 (28) severe = 4 (12%)	13 (39%)	14 (42%)	7 (21%)
**communication disorders** ***n* = 29**	5.17 ± 1.49	male = 18 (62%)	38.86 ± 7.20	35.79 ± 5.21	language disorder = 19 (66%) speech sound disorder = 9 (31%) stattering = 1 (4%)	4 (14%)	4 (14%)	0 (0%)
**attention deficit/hyperactivity disorder** ***n* = 21**	10.76 ± 3.51	male = 15 (71%)	44.33 ± 8.12	40.52 ± 6.48	combined = 15 (71%) inattentive = 3 (19) hyperactive/impulsive = 2 (10%)	5 (24%)	3 (14%)	12 (57%)
**behavioral disorders** ***n* = 16**	11.67 ± 2.58	male = 12 (75%)	42.50 ± 4.52	40.13 ± 4.40	oppositional-defiant disorder =13 (81%) conduct disorder = 3 (19%)	12 (75%)	2 (13%)	7 (44%)
**anxiety disorders** ***n* = 15**	11.97 ± 3.74	male = 7 (47%)	44.20 ± 7.57	42.07 ± 5.64	generalized anxiety = 8 (53%) school phobia = 5 (33%) social anxiety disorder = 3 (20%)	11 (73%)	2 (13%)	12 (80%)
**mood disorders** ***n* = 15**	11.73 ± 2.64	male = 6 (40%)	43.27 ± 7.46	40.93 ± 4.52	depressive disorder = 11 (69%) bipolar disorder = 4 (31%)	12 (80%)	3 (20%)	15 (100%)
**Total sample** ***n* = 369**	9.84 ± 4.29	male = 221 (60%)	43.92 ± 7.09	40.82 ± 6.29	-	122 (33%)	76 (21%)	155 (42%)

* The neuropsychiatric diagnosis was made according to the diagnostic criteria of the Diagnostic and Statistical Manual of Mental Disorders, 5th edition (DSM-5) [34].

**Table 3 ijerph-20-03924-t003:** Statistical comparison between average scores of the Child Behavior CheckList and the Parental Stress Index at Time 2 (1- ear after the pandemic) and at Time 1 (March–May 2020).

Standardized Neuropsychological Test	Time 0 (Mean ± SD)	Time 1 (Mean ± SD)	Time 2 (Mean ± SD)	ANOVA for Repeated Measures	
**Parental Stress Index (PSI)**					
Parental Distress (PD)	60.29 ± 27.27	72.38 ± 28.04	72.45 ± 28.65	F = 59.461	***p* < 0.001**
Parent-Child Difficult Interaction (P-CDI)	63.61 ± 24.29	76.82 ± 24.65	76.52 ± 25.41	F = 85.427	***p* < 0.001**
Difficult Child (DC)	63.84 ± 26.57	75.99 ± 26.47	75.76 ± 26.75	F = 58.188	***p* < 0.001**
Total Stress (TS)	63.45 ± 24.63	76.87 ± 25.94	75.68 ± 27.46	F = 91.381	***p* < 0.001**
**Child Behavior CheckList (CBCL) ** **6–18 years**					
Anxiety/Depression	59.36 ± 7.53	68.76 ± 13.18	72.73 ± 14.66	F = 152.689	***p* < 0.001**
Withdrawal/Depression	61.55 ± 8.49	69.44 ± 12.73	71.65 ± 12.54	F = 110.756	***p* < 0.001**
Somatic complaints	58.52 ± 7.97	68.59 ± 13.43	69.71 ± 13.09	F = 133.092	***p* < 0.001**
Socialization	62.20 ± 8.74	69.77 ± 12.92	70.74 ± 12.83	F = 74.966	***p* < 0.001**
Thought problems	61.40 ± 9.87	66.06 ± 11.96	66.70 ± 12.06	F = 45.245	***p* < 0.001**
Attention problems	62.75 ± 9.40	68.88 ± 11.48	69.03 ± 11.46	F = 64.286	***p* < 0.001**
Rule-breaking behavior	58.16 ± 7.02	65.28 ± 12.59	65.55 ± 12.63	F = 74.932	***p* < 0.001**
Aggressive behavior	60.80 ± 10.40	67.13 ± 13.26	67.29 ± 13.30	F = 67.049	***p* < 0.001**
Affective problems	61.87 ± 8.14	68.71 ± 11.92	72.19 ± 13.37	F = 121.542	***p* < 0.001**
Anxiety problems	61.64 ± 7.58	69.26 ± 11.95	72.86 ± 13.55	F = 122.357	***p* < 0.001**
Somatic problems	57.11 ± 8.12	64.39 ± 11.54	65.34 ± 11.55	F = 76.124	***p* < 0.001**
ADHD	60.52 ± 7.36	66.23 ± 10.07	66.63 ± 10.01	F = 68.650	***p* < 0.001**
Oppositional-defiant problems	58.01 ± 7.46	64.02 ± 10.93	64.55 ± 11.03	F = 73.968	***p* < 0.001**
Conduct problems	57.05 ± 7.03	62.92 ± 10.67	63.26 ± 10.70	F = 92.263	***p* < 0.001**
Internalizing problems	59.86 ± 9.95	68.37 ± 13.37	71.85 ± 14.54	F = 112.788	***p* < 0.001**
Externalizing problems	57.69 ± 9.72	65.83 ± 14.03	66.12 ± 14.13	F = 73.812	***p* < 0.001**
Total problems	60.41 ± 9.53	68.77 ± 13.71	70.43 ± 13.57	F = 119.146	***p* < 0.001**
**Child Behavior CheckList (CBCL) ** **1.5–5 years**					
Emotional response	54.54 ± 11.71	66.40 ± 13.63	67.11 ± 14.13	F = 13.496	***p* < 0.001**
Anxiety/Depression	55.69 ± 8.56	64.21 ± 12.73	67.16 ± 13.76	F = 20.873	***p* < 0.001**
Somatic complaints	55.03 ± 7.08	63.29 ± 12.47	65.97 ± 13.00	F = 21.449	***p* < 0.001**
Withdrawal	63.62 ± 12.94	68.56 ± 14.42	69.39 ± 14.68	F = 4.690	***p* = 0.013**
Sleep problems	56.91 ± 11.10	59.49 ± 11.49	62.98 ± 12.63	F = 5.416	***p* = 0.007**
Attention problems	61.93 ± 10.63	65.00 ± 12.35	64.55 ± 12.06	F = 2.891	***p* = 0.063**
Aggressive behavior	56.22 ± 9.32	66.34 ± 15.30	67.45 ± 14.89	F = 21.749	***p* < 0.001**
Affective problems	56.63 ± 9.00	63.19 ± 12.50	65.66 ± 13.25	F = 13.246	***p* < 0.001**
Anxiety problems	56.54 ± 8.32	63.69 ± 12.12	67.40 ± 12.55	F = 18.498	***p* < 0.001**
Pervasive problems	63.74 ± 12.16	66.29 ± 12.54	66.26 ± 11.90	F = 5.012	***p* = 0.010**
ADHD	59.26 ± 8.39	62.94 ± 9.11	63.03 ± 9.41	F = 7.006	***p* = 0.002**
Oppositional-defiant problems	59.09 ± 7.24	59.96 ± 9.76	61.05 ± 10.70	F = 9.537	***p* < 0.001**
Internalizing problems	56.71 ± 12.50	64.44 ± 15.86	67.71 ± 15.01	F = 15.948	***p* < 0.001**
Externalizing problems	56.04 ± 12.11	65.66 ± 17.03	66.56 ± 16.22	F = 14.287	***p* < 0.001**
Total problems	57.69 ± 13.55	65.56 ± 16.92	67.23 ± 15.86	F = 10.773	***p* < 0.001**

*p*-value < 0.05 are in bold.

**Table 4 ijerph-20-03924-t004:** Post-hoc analysis.

Standardized Neuropsychological Test	Time 0 vs. Time 1	Time 0 vs. Time 3	Time 2 vs. Time 3
**Parental Stress Index (PSI)**			
Parental Distress (PD)	***p* < 0.001**	***p* < 0.001**	*p* = 1.000
Parent-Child Difficult Interaction (P-CDI)	***p* < 0.001**	***p* < 0.001**	*p* = 1.000
Difficult Child (DC)	***p* < 0.001**	***p* < 0.001**	*p* = 1.000
Total Stress (TS)	***p* < 0.001**	***p* < 0.001**	*p* = 0.894
**Child Behavior CheckList (CBCL) ** **6–18 years**			
Anxiety/Depression	***p* < 0.001**	***p* < 0.001**	***p* < 0.001**
Withdrawal/Depression	***p* < 0.001**	***p* < 0.001**	***p* = 0.017**
Somatic complaints	***p* < 0.001**	***p* < 0.001**	***p* = 0.008**
Socialization	***p* < 0.001**	***p* < 0.001**	***p* = 0.005**
Thought problems	***p* < 0.001**	***p* < 0.001**	*p* = 0.068
Attention problems	***p* < 0.001**	***p* < 0.001**	*p* = 1.000
Rule-breaking behavior	***p* < 0.001**	***p* < 0.001**	*p* = 0.210
Aggressive behavior	***p* < 0.001**	***p* < 0.001**	*p* = 1.000
Affective problems	***p* < 0.001**	***p* < 0.001**	***p* < 0.001**
Anxiety problems	***p* < 0.001**	***p* < 0.001**	***p* < 0.001**
Somatic problems	***p* < 0.001**	***p* < 0.001**	***p* = 0.024**
ADHD	***p* < 0.001**	***p* < 0.001**	*p* = 0.223
Oppositional-defiant problems	***p* < 0.001**	***p* < 0.001**	***p* = 0.012**
Conduct problems	***p* < 0.001**	***p* < 0.001**	*p* = 0.347
Internalizing problems	***p* < 0.001**	***p* < 0.001**	***p* < 0.001**
Externalizing problems	***p* < 0.001**	***p* < 0.001**	*p* = 0.370
Total problems	***p* < 0.001**	***p* < 0.001**	***p* < 0.001**
**Child Behavior CheckList (CBCL) ** **1.5–5 years**			
Emotional response	***p* < 0.001**	***p* < 0.001**	*p* = 1.000
Anxiety/Depression	***p* < 0.001**	***p* < 0.001**	*p* = 0.133
Somatic complaints	***p* < 0.001**	***p* < 0.001**	***p* = 0.043**
Withdrawal	***p* = 0.010**	***p* = 0.017**	*p* = 1.000
Sleep problems	*p* = 0.751	***p* = 0.042**	***p* = 0.012**
Attention problems	*p* = 0.136	*p* = 0.411	*p* = 0.440
Aggressive behavior	***p* < 0.001**	***p* < 0.001**	*p* = 1.000
Affective problems	***p* < 0.001**	***p* < 0.001**	*p* = 0.136
Anxiety problems	***p* < 0.001**	***p* < 0.001**	***p* = 0.024**
Pervasive problems	*p* = 0.084	*p* = 0.258	*p* = 0.077
ADHD	***p* = 0.001**	***p* = 0.004**	*p* = 1.000
Oppositional-defiant problems	***p* < 0.001**	***p* < 0.001**	*p* = 0.984
Internalizing problems	***p* < 0.001**	***p* < 0.001**	*p* = 0.168
Externalizing problems	***p* < 0.001**	***p* < 0.001**	*p* = 1.000
Total problems	***p* < 0.001**	***p* < 0.001**	*p* = 0.706

*p*-value < 0.05 are in bold.

**Table 5 ijerph-20-03924-t005:** Spearman correlation analysis between Child Behavior Checklist and Parental Stress Index subscales.

			CBCL Total Problems	CBCL Externalizing Problems	CBCL Internalizing Problems
PSI/SF	Parental Distress	r	0.343	0.302	0.335
		***p*-value**	**<0.001**	**<0.001**	**<0.001**
	Parent-Child Difficult Interaction	r	0.419	0.364	0.410
		***p*-value**	**<0.001**	**<0.001**	**<0.001**
	Difficult Child	r	0.413	0.348	0.388
		***p*-value**	**<0.001**	**<0.001**	**<0.001**
	Total Stress	r	0.403	0.379	0.424
		***p*-value**	**<0.001**	**<0.001**	**<0.001**

CBCL = Child Behavior Checklist. PSI/SF = Parental Stress Index Short Form. *p*-value < 0.05 are in bold.

## Data Availability

The data presented in this study are available on request from the corresponding author.

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
