# Peer review of "COVID-19 Pandemic: 1-Year Follow-Up in Children and Adolescents with Neuropsychiatric Disorders"

_ijerph, 2023, doi:10.3390/ijerph20053924_

Round 1
Reviewer 1 Report
The authors showed the results on the long-term mental health impact of the COVID-19 pandemic on an Italian population of children and adolescents with neurological and psychiatric disorders comparing the first lock-down period with 1-year after the first lockdown (March-May 2021—Time 2), focusing on psychological, emotional, and behavioral changes and the effect on parental stress, before (Time 0), during (Time 1) and 1-year after (Time 2) the first national lockdown.
The study is very interesting, current, and well-conducted. However, some aspects could be improved.
1. it would be helpful to consider age groups with a narrower range. I do not ask the authors to run further analysis but to discuss this in the discussion.
2. The bibliography should be integrated with current research in this area.
For example Vicari S, Pontillo M. Developmental Psychopathology in the COVID-19 period. COVID-19 Pandemic Impact on Children and Adolescents' Mental Health. Psychiatr Danub. 2021 Dec;33(Suppl 11):33-35. The report of this study (or of others) could emphasize the impact of the lockdown on pathological clinical conditions.
The same concerns the studies analyzing the effects of lockdown on the quality of life of neurotypical adolescents (for example Lardone A, Sorrentino P, Giancamilli F, Palombi T, Simper T, Mandolesi L, Lucidi F, Chirico A, Galli F. Psychosocial variables and quality of life during the COVID-19 lockdown: a correlational study on a convenience sample of young Italians. PeerJ. 2020 Dec 18;8:e10611. doi: 10.7717/peerj.10611.)
3. The limitations of the study should be discussed in the Discussion by the authors
Author Response
REVIEWER #1
The authors showed the results on the long-term mental health impact of the COVID-19 pandemic on an Italian population of children and adolescents with neurological and psychiatric disorders comparing the first lock-down period with 1-year after the first lockdown (March-May 2021—Time 2), focusing on psychological, emotional, and behavioral changes and the effect on parental stress, before (Time 0), during (Time 1) and 1-year after (Time 2) the first national lockdown.
The study is very interesting, current, and well-conducted. However, some aspects could be improved.
- it would be helpful to consider age groups with a narrower range. I do not ask the authors to run further analysis but to discuss this in the discussion.
Authors response: we thank the reviewer for the suggestions. Our choice of age range was "forced" due to the structure of the CBCL itself in two form (above and under 6 years), therefore the discussion was carried out considering these two age ranges. However, we agree with the reviewer's that would have been interesting to explore also this aspect, so we intend to perform a new study dividing the sample into smaller age groups and making a new statistical comparison between them. We have specified this point in the limits and future perspectives, in the Discussion Section, as follow:
(…) Another limitation of the study is that we have considered in our analysis two large range of age (under and over 6 years), so we intend to perform a new study by dividing the sample into some smaller age groups and making a comparison between them. (…)
- The bibliography should be integrated with current research in this area. For example:
- Vicari S, Pontillo M. Developmental Psychopathology in the COVID-19 period. COVID-19 Pandemic Impact on Children and Adolescents' Mental Health. Psychiatr Danub. 2021 Dec;33(Suppl 11):33-35. The report of this study (or of others) could emphasize the impact of the lockdown on pathological clinical conditions.
- The same concerns the studies analyzing the effects of lockdown on the quality of life of neurotypicaladolescents (for example Lardone A, Sorrentino P, Giancamilli F, Palombi T, Simper T, Mandolesi L, Lucidi F, Chirico A, Galli F. Psychosocialvariables and quality of life during the COVID-19 lockdown: a correlational study on a convenience sample of youngItalians. PeerJ. 2020 Dec 18;8:e10611. doi: 10.7717/peerj.10611.)
Authors response: we thank the reviewer for the suggestions. We integrated these research in the Introduction, as follow:
(…) An Italian online survey conducted on 213 young adults during the first lockdown, suggested that social connectedness and loneliness were significant predictors of Quality of Life (QoL). The authors highlighted the importance of analyzing both social context and psychological factors in order to devise intervention strategies to improve the QoL during COVID-19 lockdown and underlined how much human relationships are fundamental for maintaining physical and psychological well-being [7] (…)
(…) In this regard, during the first and second wave of COVID-19 pandemic, the Child and Adolescence Neuropsychiatry Service at the Children Hospital Bambino Gesù in Rome has recorded a significant increase in access for mood disorders, self-injurie behaviors and suicidal ideation [17]. (…)
- The limitations of the study should be discussed in the Discussion by the author
Authors response: we thank the reviewer for the suggestions. We have discussed the limitations of the study in the Discussion section as follows:
(…) This study has several limitations, the first of which is the lack of a control group that would have allowed us to compare the evolution of emotional-behavioral symptoms in our sample and in a sample of children and adolescents without neuropsychiatric conditions. Another limitation of the study is that some social and psychological factors that could have affected the emotional-behavioral profile and parental stress (such as social connection, social support, feeling of loneliness, fear of contagion) have not been quantified. Furthermore, our study was conducted on a population homogeneous by geographical origin, therefore our conclusions are strictly related to the local conditions in the selected time (containment measures adopted, level of contagion, etc.). Another limitation of the study is that we have considered in our analysis two large range of age (under and over 6 years), so we intend to perform a new study by dividing the sample into some smaller age groups and making a comparison between them.(…)
Reviewer 2 Report
I would like to thank the editors of MDPI and the journal IJERPH for the possibility to review this work.
It is methodologically sound and has been well reviewed. The introduction makes a theoretical study of the main problems of the children studied.
The material and methods section explains the population and how the sample was obtained. The measuring instruments are adequate and well detailed.
The results detail well the data that can be obtained. However, in the statistical analysis with paired data at three points in time (T0-T1-T2), there is an error in that they only analyse the data between T1-T2, but not the evolution with repeated measures by means of a repeated measures ANOVA analysis; with intersubject and intrasubject tests. In addition, some of the data in table 3 can be presented as a figure with the evolution of the data.
This lack of analysis is a limitation of the work, which would be greatly improved by it.
The discussion is correct and debates the results, but if we had a correct statistical analysis, it would be much better and would help us to know the data better.
The bibliography is up to date and correctly written.
Author Response
REVIEWER #2
I would like to thank the editors of MDPI and the journal IJERPH for the possibility to review this work. It is methodologically sound and has been well reviewed.
The introduction makes a theoretical study of the main problems of the children studied.
The material and methods section explains the population and how the sample was obtained. The measuring instruments are adequate and well detailed.
The results detail well the data that can be obtained. However, in the statistical analysis with paired data at three points in time (T0-T1-T2), there is an error in that they only analyse the data between T1-T2, but not the evolution with repeated measures by means of a repeated measures ANOVA analysis; with intersubject and intrasubject tests. In addition, some of the data in table 3 can be presented as a figure with the evolution of the data. This lack of analysis is a limitation of the work, which would be greatly improved by it.
The discussion is correct and debates the results, but if we had a correct statistical analysis, it would be much better and would help us to know the data better.
The bibliography is up to date and correctly written.
Authors response: we thanks the reviewer for the comment and the suggestions. We performed a new statistical analysis using ANOVA for repeated measures with within-subject and inter-subject tests, as suggested by the reviewer.
The Methods section has been changed as follows:
(…) For evaluation of the changes in mean scores over time we performed the variance analysis ANOVA for repeated measures with within-subject and inter-subject test. The subsequent post-hoc analysis were performed by Bonferroni test. (…)
The Results section has been changed, as follows:
(…) The analysis of variance ANOVA for repeated measures showed significant changes over time in all the subscale of PSI and CBCL 6-18; in the CBCL 1.5-5 significant changes were recorder in all the subscales considered, except for attention problems subscale (p=0.063).
The post-hoc analysis showed that the comparison between mean PSI scores during the first lockdown (Time1) and 1-year after the pandemic (Time2) showed no significant changes in all subscales analyzed in the total sample (p>0.05 in all the PSI subscales). The mean scores of the following subscales of CBCL 6-18 were statistically increased at Time2 compared to Time1: Anxiety/Depression (p<0.001), Withdrawal/Depression (p=0.017), Somatic complaints (p=0.008), Socialization (p=0.005), Affective problems (p<0.001), Anxiety problems (p<0.001), Somatic problems (p=0.024), Oppositional-defiant problems (p=0.012), Internalizing problems (p<0.001), Total problems (p<0.001). We not found significant changes in the remain subscales. The mean scores of the following subscales of CBCL 1.5-5 were statistically increased at Time2 compared to Time1: Somatic complaints (p=0.043), Sleep problems (p=0.012), Anxiety problems (p=0.024). (…)
We also modified Table 3 (ANOVA results) and we added Table 4 (Post-Hoc results).
As suggested we added Figure 1 that summarizes the most significant results.
Since the results remained almost unchanged in terms of significance the Discussion section was only slightly modified as follows:
(…)More in details, regarding the CBCL 6-18, the statistical analysis of the changes in the mean scores over time showed that there was a significant worsening in all subscales analyzed during the first lockdown compared to the pre-pandemic period, furthermore, after 1-year follow-up, the following scales were further worsened: Anxiety/Depression, Withdrawal/Depression, Somatic complaints, Socialization, Affective problems, Anxiety problems, Somatic problems, Oppositional-defiant problems, Internalizing problems, Total problems. (…)
(…) Regarding the CBCL 1.5-5, the analysis of the changes over time showed a significant worsening in all subscales during the first lockdown compared to the pre-pandemic period, while, after 1-year follow-up, the following scales were further worsened : Somatic complaints, Anxiety problems, and Sleep problems (…)
(…) However, despite the lack of other data in the literature, our experience suggests that anxiety symptoms and somatization are still high after one year of the pandemic even in younger children. In the sample of children under the age of 6, sleep disturbances had also worsened further, probably due to the change in habits and anxious symptoms. (…)
(…)The analysis of change in parental stress over time showed a significant increase in parental stress levels during the first lockdown compared to pre-COVID levels; on the contrary, after 1-year follow-up no significant change in parental stress was found. This could suggest that caregivers of patients with neuropsychiatric diagnosis were already stressed during the first lockdown and are maintaining constant their stress levels. (…)
Round 2
Reviewer 2 Report
The authors have made the necessary changes so that the paper can be accepted.